# A Semi Empirical Regression Model for Critical Dent Depth of Externally Corroded X65 Gas Pipeline

**DOI:** 10.3390/ma15165492

**Published:** 2022-08-10

**Authors:** Yue Yang, Xiaoben Liu, Hong Yang, Weilun Fang, Pengchao Chen, Rui Li, Hui Gao, Hong Zhang

**Affiliations:** 1National Engineering Laboratory for Pipeline Safety/MOE Key Laboratory of Petroleum Engineering/Beijing Key Laboratory of Urban Oil and Gas Distribution Technology, China University of Petroleum-Beijing, Beijing 102249, China; 2Beijing Key Laboratory of Pipeline Critical Technology and Equipment for Deepwater Oil & Gas Development, Beijing Institute of Petrochemical Technology, Beijing 102627, China; 3China Energy Engineering Group Zhejiang Electric Power Design Institute Co., Ltd., Hangzhou 310012, China; 4PipeChina Institute of Science and Technology, Langfang 065000, China

**Keywords:** corrosion dent, critical dent depth, finite element analysis, failure analysis

## Abstract

External corrosion dent is a common type of compound dent. On the one hand, this type of compound dent reduces the bearing area and bearing capacity of the pipeline. On the other hand, it leads to an increase in the stress–strain concentration in the dent and reduces the anti-fatigue load capacity of the pipeline, which is more harmful to the service safety of the pipeline than the simple dent. In this study, the reliability of the modeling method was verified by the numerical inversion of the full-size dented pipe test. A three-dimensional finite element model for a pipe with a small corrosion dent was established by analyzing the internal detection data on corrosion defects of pipes with a diameter of 813 mm. The failure criterion of the corrosion dent pipe and the calculation method of the critical dent depth were determined. The influence of corrosion depth, length, width, internal pressure load, curvature radius of indenter, and diameter–thickness pipeline ratio on critical dent depth was investigated. Finally, a critical dent-depth prediction formula was developed based on the numerical results. This study provides a reference and significant guidance for the applicability evaluation of corroded sunken pipelines.

## 1. Introduction

Since the Second World War, oil and natural gas have become an important strategic resource and emerged as a global asset for the development and maintenance of national security and stability. At the same time, oil and gas pipelines, as the most economical, safe, and environmentally friendly candidates for transporting oil and natural gas, have also formed an essential infrastructure for the development of the energy industry in various countries [1].

Dent is a type of typical mechanical damage defect in pipelines, which is a permanent inward plastic deformation in the pipe wall [2]. Dents generate local stress and strain concentrations in the pipeline and seriously threaten its integrity. Although the relevant blast experiment results showed that a simple, smooth dent does not significantly reduce the blasting intensity of the pipeline [3,4,5,6], a large number of full-size fatigue tests [5,6,7] have revealed that the stress and strain concentration caused by dents led to a reduction in the fatigue life of pipelines. In particular, for unconstrained dents, frequent fluctuations in internal pressure cause frequent concaves and rebounds in the dent area and lead to local fatigue failure. Moreover, excessive deep dents not only hinder the transition of internal detector, pipeline scraper, and other equipment, resulting in blockage, but are prone to crack initiation [8], which is expected to become a major hidden danger in the future operation of pipelines. Some sharper dents even undergo a perforation phenomenon, directly leading to leakages in the transport medium. Moreover, this situation damages the environment, and fires and explosions may occur, resulting in serious accidents and casualties.

Compound dents formed by the combination of simple dents and other defects are also common in pipelines. For example, some excavation equipment in the construction process may hit the pipeline and form a dent and simultaneously scratch the pipeline. Under the combined influence of the hard material at the bottom of the ditch, leading to damage to the outer coating and corrosion during operation, a corrosion dent can also form in the pipeline. The compound dent reduces the bearing area and bearing ability of the pipeline. It also leads to an increase in the stress concentration in the dent and reduces the anti-fatigue load ability of the pipeline, which is more harmful to the service safety of the pipeline than a simple dent.

Recently, a large number of experiments and numerical simulation studies have been performed on corroded pipelines. ASME B31G [9], DNV RP F101 [10], PCORRC method [11], and other similar methods are standard specifications and methods to evaluate the residual strength of corroded pipelines formed on this basis. However, these specifications and methods are only used to explore the corrosion defects, and a detailed and reasonable applicability evaluation method for corrosion dents has not been reported to date. Dinovitzer et al. [12] established an evaluation model for pipelines with dent based on the crack propagation principle. The influence of corrosion depth, corrosion form and corrosion location, and the relative time of corrosion and dent formation, on the residual life of the pipeline was analyzed using a nonlinear finite element method (FEM). For example, Liu et al. [13] studied the influence of corrosion size, diameter of discharge head, pressing depth, and internal pressure on critical internal pressure load of a pipeline through FE elastoplastic analysis. Based on the PCORRC method, the basic multi-factor failure evaluation formula was derived, and the undetermined coefficients in the formula were fitted using nonlinear regression analysis. Ma et al. [14] studied the stress and strain distribution of pressure pipelines with internal corrosion dents with FEM. Tian and Zhang [15] studied the failure mechanism and internal pressure limits of scratch dent through experiments and FEM, and proposed a failure criterion based on equivalent plastic strain.

To date, studies on corrosion dents in the evaluation of compound dents have not been extensively explored. Most of the evaluation methods reported in corrosion dent studies can only be applied to pipelines with specific sizes and defect combinations, and thus lack generality. In most studies, the size of the corrosion defect on the corrosion dents is too large, while the actual corrosion dents are relatively small. Moreover, the corroded pipeline model, established using the FE numerical simulation, does not chamfer the edge in the corrosion pit, which easily causes local stress and strain concentrations at the edge, making the results inaccurate.

In this study, based on the analysis of the internal inspection data of a pipeline with small corroded dents, an FEM of a pipeline with small corroded dents was established and grid sensitivity was analysed. The model was parameterized using the Python language, and the failure criterion and calculation method for the critical dent depth were discussed and determined. Based on the parametric FEM and numerical calculation results, the influence of corrosion depth, length, width, internal pressure load, curvature radius of the indenter, and diameter–thickness ratio of the pipeline on critical dent depth was analyzed. Finally, the basic form of the critical dent depth prediction formula was quantitatively constructed according to Pi. Based on the FEM calculation results of 453 groups of different working conditions, the nonlinear fitting software 1stOpt was used to fit the undetermined coefficients in the formula, and a good prediction effect was obtained.

## 2. Detection of a Pipeline with External Corrosion Dent

Notably, external corrosion dents are common compound dents. The dent itself leads to the concentration of stress and strain in the pipeline, and the geometric discontinuity of the pipeline, caused by corrosion, aggravates the stress and strain concentration, which increases the possibility of stress corrosion cracking in the pipeline. Moreover, wall thinning caused by corrosion directly reduces the ultimate internal pressure in the pipeline. Two common occurrences are responsible for the generation of corrosion dents. First, when the external anticorrosive coating of the pipeline is destroyed by rocks at the bottom of the pipeline or mechanical excavation, this leads to the direct exposure of the pipeline body to the external environment. In the subsequent operating phase, failures in cathodic protection lead to progressive corrosion dents. Second, corrosion occurs in the pipeline during operation, and then a dent is formed based on the corrosion due to the excavating operation, and for other reasons. Notably, under the conditions of the same-size corrosion and dent, the second case is more dangerous than the first one.

The magnetic flux leakage testing data of a pipeline obtained from 2011 to 2012 indicate the presence of 8131 external metal loss defects in the pipeline, with an average of 8.76 defects per kilometer, with a maximum defect depth of 35% of the wall thickness. These metal loss defects mainly include manufacturing defects (about 29.6%) and corrosion defects (about 70.4%). According to the external metal loss defect size distribution of a pipeline (Figure 1), most of the metal losses were defects with a small length and width, and the defects with a metal loss depth of less than 10% of the wall thickness accounted for nearly 80%. Moreover, 62 corrosion dents were detected during internal inspection, all of which corresponded to low external corrosion, with a maximum dent depth of 5.38% of the outer diameter *OD* and a maximum corrosion depth of 20% of the wall thickness.

## 3. Full-Size Dented Pipe Test and Numerical Inversion

In this paper, a dented pipeline model was established based on the full-size dented pipe test carried out by Rafi [16] to carry out the inversion and verify the accuracy of the finite element modeling method.

### 3.1. Overview of the Full-Size Dented Pipe Test

The outer diameter *OD* of the test pipe was 274 mm, the wall thickness was 8.2 mm, and the length was 1100 mm. The ends of the pipe were sealed with end covers. The pipe was API 5L X52 pipeline steel, and its mechanical properties are shown in Table 1 and Figure 2.

During the test, an internal pressure of 4.83 MPa was applied to the pipe through a static hydraulic pump and a hydraulic press connected to a semi-spherical indenter with a curvature radius of 25 mm (Figure 3a). This was used to apply an external load on the middle of the pipe top to form a dent. The process of forming the dent was divided into four loading steps. After each loading step, the pressure head was removed and the internal pressure was unloaded, and the depth of the dent after rebound was measured after four loading steps: 3.3% *OD*, 4.7% *OD*, 6.2% *OD*, and 8% *OD*, respectively. The strain near the outer wall of the dent was measured by a pre-arranged strain gauges, as shown in Figure 3c,d.

### 3.2. Finite Element Model of Full-Size Dented Pipeline

A dented pipe model was established in nonlinear finite element software ABAQUS^®^ and the above test was inverted. According to the symmetry of the model, the 1/4 model of the pipeline was established by using a four-node reduced integral shell element (S4R), and the semi-spherical indenter model was established using a discrete rigid body. The size of the model was consistent with the test. According to Formula (1), the pipe engineering stress–strain data were converted into true stress–strain data and set in the model.
(1)$$\begin{array}{l} \varepsilon_{true}=\ln\left(1+\varepsilon_{nom}\right)\\ \sigma_{true}=\sigma_{nom}\left(1+\varepsilon_{nom}\right) \end{array}$$
where *ε_true_* and *ε_nom_* are the true strain and engineering strain; *σ_true_* and *σ_nom_* are the true stress and engineering stress (MPa).

Symmetric boundary conditions were set to the circumferential cross-section and the axial cross-section at the pressing position under the indenter. The axial displacement of the cross-section at the far end of the pipe was restricted and the pipe bottom axis was fixed. The area that would be a dent was divided into dense mesh with 2 mm × 2 mm size, and other parts of the mesh were sparsely arranged to improve computational efficiency (Figure 4).

The whole numerical simulation process was divided into four loading and unloading processes. Each loading and unloading process included four analysis steps: internal pressure application, head loading, head unloading, and internal pressure unloading. The loading and unloading processes were different only in head loading and head unloading, and the loading and unloading of internal pressure were the same. The displacement of reference points was set on the indenter to control the loading and unloading of the indenter. According to the displacement load curve of the indenter in the test, the loading displacements of the indenter were set four times, as 14 mm, 19 mm, 24 mm, and 28 mm in the model, respectively.

### 3.3. Numerical Inversion Results and Comparative Analysis

To verify the correctness of the finite element model, the load–displacement curves during the loading and unloading processes were compared between the inversion results and the test results, as well as the circumferential strain along Line 1 and the axial strain along Line 2 in Figure 3d. From the load–displacement curves (Figure 5) and the comparison of circumferential and axial strains of the outer wall of the pipe (Figure 6), the inversion results were found to be in good agreement with the test results.

## 4. Finite Element Model of Pipeline with External Corrosion Dent

### 4.1. Overview of the Model

The pipeline model with external corrosion was established using a three-dimensional (3D), 8-node, hexahedron, quadratic, complete integral element (C3D8). According to the symmetry of structure and load, the 1/4 analysis model was established to improve the calculation efficiency. The semi-spherical indenter model was established using a rigid analytical body. The pipe diameter of the model is 813 mm, and the wall thickness ranges from 12.5 mm to 17.5 mm. According to the Saint-Venant principle [17], to avoid the influence of boundary conditions, the length between the center of the dent and the end of the pipeline should be four times the length of the outer diameter, as illustrated in Figure 7. A frictionless contact was defined between the indenter and the outer surface of the pipe.

### 4.2. Material Properties

In the FE numerical simulation, it is necessary to use the true stress–strain curve of the material, which can be obtained by converting the engineering stress–strain curve obtained by the axial tensile test. The Ramberg–Osgood model [18] can also be used to express the stress–strain relationship, and the formula in CSA-Z662 [19] is represented as follows:(2)$$\varepsilon =\frac{\sigma }{E}+\alpha \frac{\sigma }{E}{\left(\frac{\sigma }{\sigma_{s}}\right)}^{n}$$
where *E* is the elastic modulus (MPa), *α* is yield offset; *σ_s_* is the yield strength (MPa), and *n* is the strain-hardening parameter.

In this study, X65 pipeline steel was selected as the defining material of the pipeline and its specific mechanical parameters are listed in Table 2. The true stress–strain curve of the X65 material is displayed in Figure 8.

### 4.3. External Corrosion Defect

The external corrosion defects considered herein are assumed to be regular rectangles, and the size of the corrosion defects is described in terms of three indexes, namely, the axial length and the circumferential width and depth along the pipeline, as illustrated in Figure 9. The axial length of the corrosion defect is represented by $L\sqrt{Dt}$ (*L* is the corrosion length coefficient, *D* is the outer diameter of the pipe and *t* is the wall thickness); the circumferential width is represented by $W\sqrt{Dt}$ (*W* is the corrosion width coefficient); the corrosion depth is represented by *d*. In order to reduce the distortion caused by geometric discontinuity of the edge of the corrosion defect in the deformation process, chamfering is performed on the edge of the corrosion pit, as presented in Figure 10.

### 4.4. Analysis of Step and Load

The entire analysis process is divided into four steps, as presented in Figure 11. First, the internal surface of the pipeline is defined, and the internal pressure load is applied. In the second step, a small radial displacement load pointing to the central axis of the pipeline is applied to the reference point of the indenter in a cylindrical coordinate system to establish contact between the indenter and the pipeline. In this process, contact stability control is used to resist rigid body displacement until contact is completely established. Then, the full displacement load is applied to the reference point of the indenter, to significantly hollow the pipe. Finally, a displacement load is applied in the opposite direction to the indenter to make it leave the surface of the pipeline, and the pipeline with defect loses its external load to complete the rebound. The Nlgeom option is enabled in the entire process to ensure the convergence and accuracy of the model’s nonlinear response under large deformation conditions.

### 4.5. Boundary Condition and Mesh Generation

Symmetry constraints were set for both circumferential section A and axial section B. Section A constrains the displacement degrees of freedom (dof) U3 of axial displacement and the rotational dof UR1 and UR2 of the other two directions. Section B constrains the displacement dof U1 in the *X* direction and the rotational dof UR2 and UR3 in the other two directions. For the end section C, only the axial displacement dof U3 is constrained. Moreover, the dofs of six directions of the tube base line *l* are constrained. The FE mesh in the local region of the corrosion defect location is illustrated in Figure 12. The corrosion defect and dent deformation area are the stress and strain concentration area of the pipeline; thus, the grid in this part is encrypted. For other regions, the grid is thinned appropriately, according to the distance from defects, to improve the calculation rate.

In order to ensure the accuracy and reliability of the FEM calculation results and improve the calculation efficiency, five grid sizes were selected to conduct a grid sensitivity analysis of the model. The mesh sizes used in the sensitivity analysis are listed in Table 3.

Through simulation, the equivalent plastic strain (PEEQ) of the inner surface of the corrosion defect center (Figure 13) was extracted for comparative analysis. It was revealed that when the mesh size was less than 2 mm × 2 mm, the results did not significantly change. The mesh with dimension of 1 mm × 1 mm was too dense; thus, the calculation cost of the simulation analysis was too high. Therefore, the 2 mm × 2 mm mesh was selected as the defect site mesh of the final FE simulation.

## 5. Failure Criterion and Critical Dent Depth

### 5.1. Failure Criterion

To date, many failure criteria have been proposed in pipeline failure research. The most common failure criteria include the critical load of pipelines based on the load–displacement curve, the stress-based criterion, and the strain-based criterion. The criteria for judging pipeline failure according to the load–displacement curve include the twice elastic slope criterion [20], zero-curvature criterion [21], etc. The Rresca equivalent stress criterion and Mises equivalent stress criterion are commonly used for failure criteria based on stress. Under the action of the external load, the pipeline is in the multiaxial stress state; thus, the pipeline failure can be judged by comparing the equivalent stress and the critical stress.

It is highly desirable and urgent to determine the critical stress according to the failure mechanism of the piping material. In general, the failure of low-strength steel is based on the fracture mechanism, and mainly on the yield strength of the material; in contrast, the failure of high-strength steel is based on the plastic instability, and mainly on the tensile strength of the material. Moreover, the flow stress is considered as the judging index by the net section failure criterion. According to this criterion, when the equivalent stress at an arbitrary point along the wall thickness direction of the dangerous area of the pipeline reaches the flow stress, the pipeline fails. According to the difference in pipeline and temperature, ASME B31G [9] provides the calculation formula for flow stress.

Notably, the criterion based on stress is usually suitable for small pipeline deformations, and not for large-displacement plastic deformations, such as depressed deformation. In the case of large-displacement plastic deformation, the failure criterion based on strain, which considers that failure occurs when the reference strain reaches the critical strain, is usually more accurate. The commonly used strain-based failure criteria at present include ductile fracture damage criterion [22], strain limit damage criterion [23], minimal elongation criterion [24,25], and FE failure criterion [15,26,27,28,29,30]. According to the ductile fracture damage criterion [22], pipe failure occurs when the ductile damage index of the material reaches 1. The strain limit damage criterion is a method recommended in ASME BPVC [23] to estimate the accumulated damage as the strain limit damage by elastic-plastic FE analysis. The minimum elongation criterion is introduced in ASME B31.8 [25] to evaluate the pipeline strain level of according to the minimum material elongation in the evaluation of smooth dents.

In this study, the failure criterion based on the plastic strain of fracture, which was validated by experimental studies [15,29,30], was adopted as the failure criterion of pipelines with corrosion dents. The failure criterion based on fracture plastic strain was obtained by comparing the test values of the critical tensile load of a CT80 steel plate with the predicted values of four common failure criteria. The experimental results showed that the failure criterion based on a fracture plastic strain is more accurate; thus, the pipeline failure is determined when the equivalent plastic strain of the integral point at an arbitrary point in the FE analysis reaches the fracture strain of the material.

### 5.2. Critical Dent Depth

First, according to the mechanical parameters of the material presented in Table 2, the fracture strain of X65 pipeline was calculated to be about 0.2059. FE analysis was used to determine the first failure element of corroded pipeline in the process of depressed deformation (that is, when the equivalent plastic strain at any integral point of the model reaches 0.2059), which obtained the curve of equivalent plastic strain of the first failure element with dent depth. According to the fracture strain, the dent depth when the pipeline fails is determined as the critical dent depth *d_cr_*, and the more accurate *d_cr_* can be further determined by the interpolation method. Figure 14 illustrates that the FEM adopts a quadratic complete integral element with eight nodes; therefore, each element has eight integral points. The *d_cr_* is determined by selecting the integral curve that reaches the fracture strain first.

## 6. Numerical Results and Discussion

During the formation of external corrosion dents in the pipeline, different mechanical responses are caused under the different conditions of defects and loads. This section analyzes the parameters, mainly including defect geometric parameters (depth *d*, length $L\sqrt{Dt}$, width of corrosion $W\sqrt{Dt}$), load parameters (curvature radius of indenter *R*, internal pressure load *P*) and pipeline parameters (diameter–thickness ratio *OD*/*t*) that may affect the critical dent depth *d_cr_* in corrosion dent pipelines. Python programming was used to parameterize the established FEM of pipeline with external corrosion dent, to simplify the parameterization calculation process and improve the analysis efficiency.

### 6.1. Influence of Defect Geometrical Parameters on Critical Dent Depth

Based on the analysis of the influence of *d* on *d_cr_*, the values of the modeling parameters were obtained and are presented in Table 4. In order to more intuitively reflect and analyze the influence of *d* on *d_cr_*, the curve of the critical dent depth to diameter *d_cr_*/*OD* ratio, changing with the corrosion depth to wall thickness *d*/*t* ratio was plotted under different *P* conditions (Figure 15).

The analyzed calculation conditions, overall, indicate that the *d* and the *d_cr_* are negatively correlated. However, when the *d* is small, the two are not completely negatively correlated. When *d* > 0.2 *t*, the *d_cr_* decreases with the increase in *d*. However, when *d* < 0.2 *t*, the relationship between them becomes complicated, but the *d* exhibits little influence on the *d_cr_*. Moreover, with the increase in *R*, the influence of the *d* on the *d_cr_* becomes more obvious.

In order to explore the role of *R* in the influence of *d* on the *d_cr_*, a *P* = 10 MPa condition was considered as an example to draw the curve of the change in *d_cr_*/*OD* with *d*/*t* under the conditions of different curvature radius ratios of indenter to diameter *R*/*OD*, as shown in Figure 16. With the increase in *d*, the influence of the increase in *R* on the *d_cr_* decreases.

The analysis of the influence of corrosion length on the *d_cr_* indicates that the *W* = 0.352, and the *L* is between 0.2 and 0.8, with an interval of 0.05. There are 13 groups of calculation conditions. The analysis results of the influence of corrosion width on the *d_cr_* reveal that the *L* = 0.4, and the *W* is in the range from 0.211 to 0.739, with an interval of 0.0352, and there are 16 groups of calculation conditions. The *d* = 0.3 *t*, *R*/*OD* = 0.08, and other parameters are the same as those in the influence analysis of the *d*. Based on the simulation calculation results, the curves in *d_cr_*/*OD*, changing with *L* and *W* under different *P*, were extracted and drawn, as shown in Figure 17.

Notably, the influence of corrosion length and corrosion width on the *d_cr_* is similar. Under the analyzed calculation conditions, when the *L* = 0.4 and the *W* = 0.352 ($L\sqrt{Dt}$ = 40.32 mm, $W\sqrt{Dt}$ = 35.47 mm), the *d_cr_* is the largest, and other length–width combinations are more dangerous. For small-size corrosion, when the length and width are close, both the length and width edges of the corrosion can contact the indenter, whose *R* = 65.04 mm at the initial contact stage, in a very short, semi-spherical indenter displacement. Owing to the special shell surface structure of the pipeline, when the annular width of the corrosion is slightly shorter than the axial length, the semi-spherical indenter can just contact the length and width edge of the corrosion at the same time, and the structure can bear the maximum external load of the indenter. Furthermore, Figure 17b illustrates that Point A (the combination of *L* = 0.4 and *W* = 0.352, i.e., $L\sqrt{Dt}$ = 40.32 mm, $W\sqrt{Dt}$ = 35.47 mm) is closer to the square combination and has a larger *d_cr_* than Point B (*L* = 0.4 and *W* = 0.387, i.e., $L\sqrt{Dt}$ = 40.32 mm, $W\sqrt{Dt}$ = 39.02 mm).

### 6.2. Influence of Load Parameters on Critical Dent Depth

The influence of load parameters include an internal pressure load and external load from the indenter. Table 5 presents the modeling parameters. In this study, the indenter is spherical in shape; thus, the influence of the external load on the indenter was simplified to the influence of the sharpness of the indenter, that is, only the curvature radius of the indenter *R* was considered.

When the *R*/*OD* = 0.08 and 0.12, respectively, the curve of the *d_cr_*/*OD,* which changes with the *P* under different *OD*/t, is shown in Figure 18.

The increase in *P* leads to a reduction in the corroded pipeline’s ability to bear the external load of the indenter head. With the increase in *P*, the *d_cr_* linearly decreases. According to the results of the action of the two indenters in the figure, when the *OD*/*t* = 65.04, the *d_cr_* of the sharper indenter case decreases from 4.43% *OD* at 4 MPa to 3.71% *OD* at 10 MPa with the change in *P*, decreasing by 19.41%. The *d_cr_* of the smoother indenter decreases from 5.73% *OD* to 4.65% *OD*, decreasing by 23.23%. Therefore, the depressurization operation can improve the failure prevention of corroded dent pipelines, and for smoother dents, the depressurization operation can improve the prevention effect more than it can for sharper dents.

In the analysis of the curve of *d_cr_*/*OD*, which changes with the *R*/*t* under different *P* (Figure 19), the *d_cr_* obviously increases with the increase in *R*. The smaller the head curvature radius, the sharper the indenter; when the indenter interacts with the pipeline, the local deformation of the pipeline is likely to be larger, which leads to a more serious local strain concentration. Therefore, it is not ideal to only consider the corrosion size and dent depth as the evaluation reference in some existing evaluation standards of corrosion dent pipeline.

Moreover, a comparative analysis of the results of pipes with different *OD*/*t* indicates that, with the increase in *OD*/*t*, the curve range of the four different internal pressures becomes wider and wider, demonstrating that the *P* has an increasingly significant influence on the *d_cr_*.

### 6.3. Influence of Diameter–Thickness Ratios on Critical Dent Depth

The change curve of *d_cr_*/*OD* with *OD*/*t* of pipeline under different *P* (Figure 20) indicates that the *d_cr_* is negatively correlated with *OD*/*t*. The *d_cr_* decreases with the increase in *OD*/*t*, and the influence of the *OD*/*t* becomes more significant with the increase in *R*. Moreover, the influence of the *P* becomes more obvious with the increase in *OD*/*t*. For the same pipeline, the smaller the *t*, the greater the prevention benefits of the depressurization operation.

## 7. Prediction Formula of Critical Dent Depth

According to the function relationship between each parameter and the intuitive analysis of the critical dent depth *d_cr_* based on the numerical results and discussion, the representative mathematical expression is allocated to each parameter accordingly, and the *d_cr_* prediction formula is constructed based on the “Pi theorem” [31,32]. The numerical results and discussion also present an important relationship between the parameters, which can be incorporated into the variable functions of each equation. Each equation is based on the concept of direct combinatorial multiplication, simply multiplying the functions of the variables as follows:(3)$$d_{cr}=f_{cr}\left(\pi_{1},\pi_{2},\pi_{3},\pi_{4},\pi_{5},\pi_{6}\right)=f_{1}\cdot f_{2}\cdot f_{3}\cdot f_{4}\cdot f_{5}\cdot f_{6}$$
where *d_cr_* is the critical dent depth; *f_cr_* is the prediction formula of critical dent depth; *π*_1_, *π*_2_, *π*_3_, *π*_4_, *π*_5_, and *π*_6_ are the dimensionless parameters affecting the critical dent depth, *π*_1_ is the corrosion depth *d*, *π*_2_ is the corrosion length coefficient *L*, *π*_3_ is the corrosion width coefficient *W*, *π*_4_ is the internal pressure coefficient *P*/*P_y_*, *P_y_* = 2*tσ_s_*/*d*, *π*_5_ is the curvature radius of the pressure head *R*, *π*_6_ is the diameter–thickness ratio *OD*/*t*; *f*_1_ is the corrosion depth function, *f*_2_ is the corrosion length function, *f*_3_ is the corrosion width function, *f*_4_ is the internal pressure function, *f*_5_ is the curvature radius function of the head, and *f*_6_ is the radial thickness ratio function.

According to the analysis of influencing factors, the basic form of function of each variable is determined as follows:(4)$$\begin{array}{l} f_{1}=\left(a_{1}\pi_{1}{}^{2}+b_{1}\pi_{1}+c_{1}\right)\pi_{2}{}^{d_{1}}\pi_{3}{}^{e_{1}}\\ f_{2}=\left(a_{2}\pi_{2}{}^{2}+b_{2}\pi_{2}+c_{2}\right)\\ f_{3}=\left(a_{3}\pi_{3}{}^{2}+b_{3}\pi_{3}+c_{3}\right)\\ f_{4}=\left(a_{4}+b_{4}\pi_{4}{}^{c_{4}}\right)\pi_{6}{}^{d_{4}}\\ f_{5}=\left(a_{5}+b_{5}\pi_{5}{}^{c_{5}}\right)\pi_{1}{}^{d_{5}}\\ f_{6}=a_{6}+b_{6}\pi_{6}{}^{c_{6}} \end{array}$$

Based on the numerical results of FE calculation of 453 corroded dents, the nonlinear fitting software 1stOpt was used to fit each coefficient in the prediction formula of critical depression depths, and the correlation coefficient was found to be 0.9557. The fitting results are presented in Table 6 and Figure 21.

The applicable ranges of all dimensionless parameters in the *d_cr_* prediction formula obtained in this study are listed in Table 7. The results of the comparative analysis show that the error between the calculation results of most prediction formulas and the FE results is less than 10%, and the prediction effect is good, which can provide a certain reference value for pipeline corrosion dent engineering evaluation.

## 8. Conclusions

In this study, the reliability of the modeling method was verified by a numerical inversion of the full-size dented pipe test. The finite element model and the value range of modeling parameters of pipe with small-size corrosion dents were established by analyzing the internal detection data of corrosion defects of an 813 mm diameter pipe. The failure criterion of the corrosion dent pipe and the calculation method of the critical dent depth were discussed and determined. The influence of corrosion depth, length, width, internal pressure load, curvature radius of indenter, and diameter–thickness ratio of the pipeline on critical dent depth was systematically investigated. Finally, a critical dent depth prediction formula was developed based on the numerical results. The following conclusions are drawn from this study:(1)When the corrosion depth is greater than 0.2 *t*, the critical dent depth decreases with the increase in corrosion depth. In contrast, for a corrosion depth of less than 0.2 *t*, the influence of the corrosion depth is small. The influence of corrosion depth on critical dent depth becomes more obvious with the increase in head curvature radius.(2)In the case of small-size corrosion dents, the closer the corrosion length and width, the greater the critical dent depth, and the greater the pipeline’s ability to bear the external load of the semi-spherical indenter.(3)Both the increase in internal pressure load and decrease in the curvature radius of indenter led to a decrease in critical dent depth and the corroded pipeline’s ability to bear the external load of the semi-spherical indenter. With the increase in internal pressure from 4 MPa to 10 MPa, the critical dent depth of the sharper and smoother indenters decrease by 19.41% and 23.23%, respectively.(4)The critical dent depth decreases with the increase in the diameter–thickness ratio, and the influence of the diameter–thickness ratio becomes more significant with the increase in the curvature radius and internal pressure.(5)In this study, a critical dent depth prediction formula was developed for an 813 mm diameter pipe with corrosion defects. A good agreement is achieved between the proposed method and the numerical results.

However, undeniably, more systematic explorations are still needed to investigate various other numerical simulation conditions of external corrosion dent pipes; the simulation conditions of the curvature radius of the indenter particularly require further attention. In this study, the sharper condition was not sufficiently evaluated. Moreover, considering only the semi-spherical indenter is not consistent with the fact that indenters with different shapes are available, which also lead to the formation of dents. Furthermore, the dent set in this study was located in the center of the corrosion defect, and the dents in other relative positions were not considered. The above deficiencies require further systematic research.

## Figures and Tables

**Figure 1 materials-15-05492-f001:**
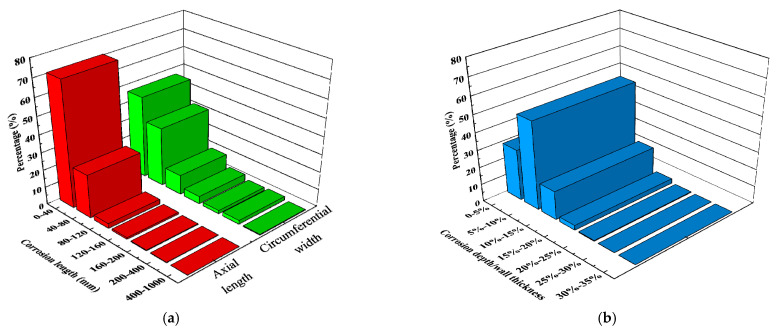
The external metal loss defect size distribution of a pipeline: (**a**) corrosion length, (**b**) corrosion depth.

**Figure 2 materials-15-05492-f002:**
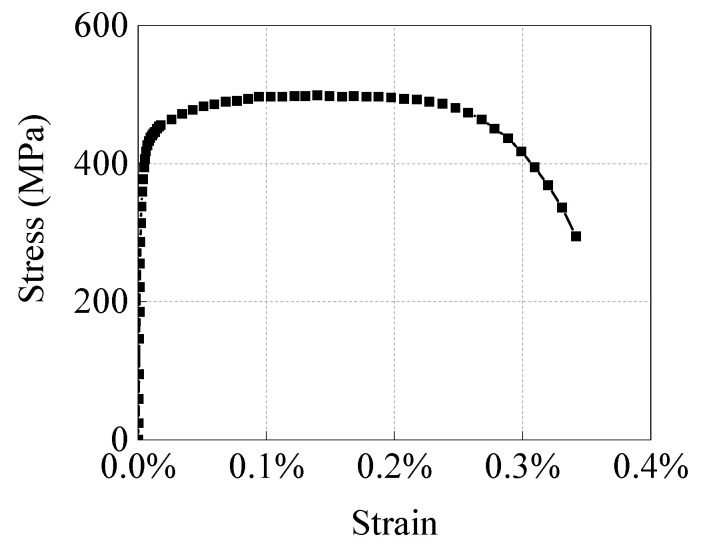
Engineering strain–stress curve of pipe for testing.

**Figure 3 materials-15-05492-f003:**
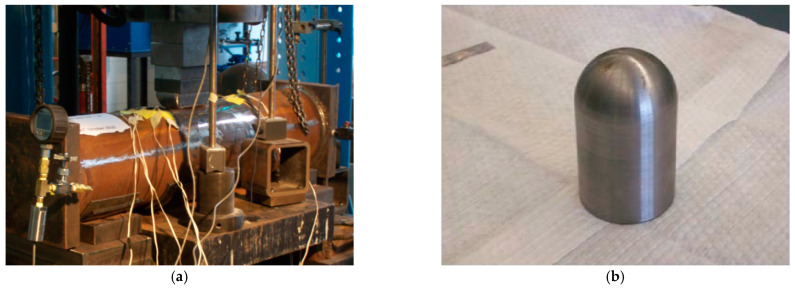
The dented pipe test photo and strain gauges arrangement: (**a**) test equipment, (**b**) semi-spherical indenter, (**c**) dented pipe and strain gauge arrangement, (**d**) schematic diagram of strain gauge arrangement [16].

**Figure 4 materials-15-05492-f004:**
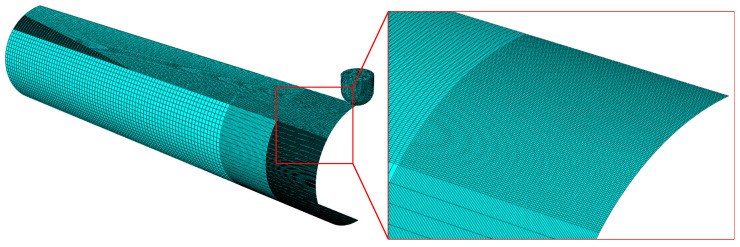
Mesh generation of the dented pipe model.

**Figure 5 materials-15-05492-f005:**
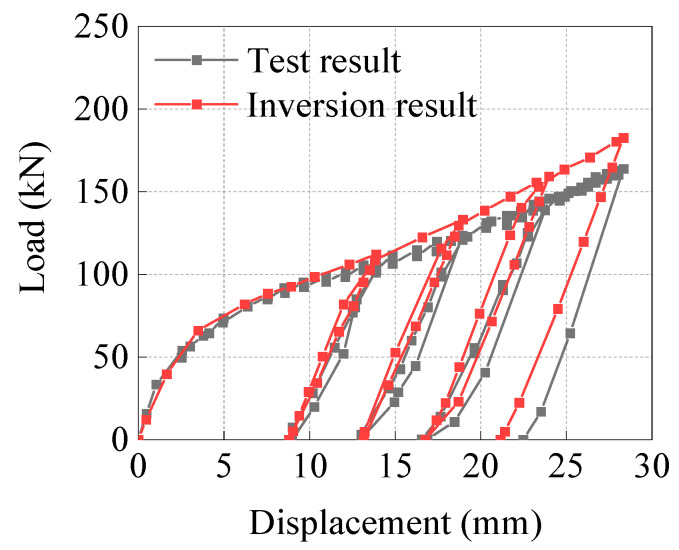
Comparison of load–displacement curves.

**Figure 6 materials-15-05492-f006:**
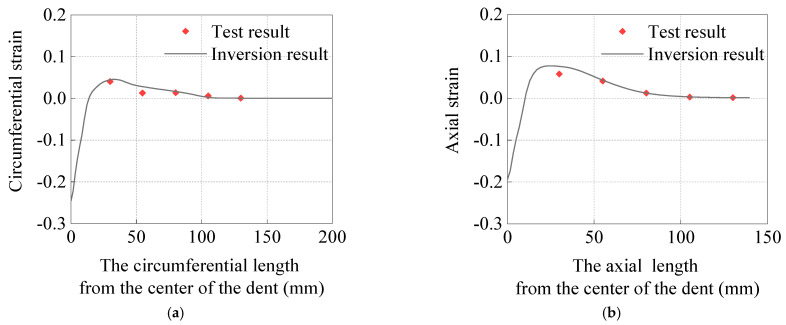
Comparison of strain results: (**a**) circumferential strain, (**b**) axial strain.

**Figure 7 materials-15-05492-f007:**
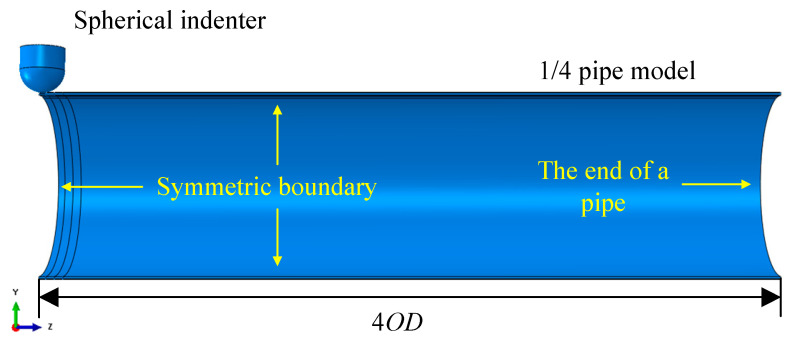
Geometric model of 1/4 pipe with external corrosion.

**Figure 8 materials-15-05492-f008:**
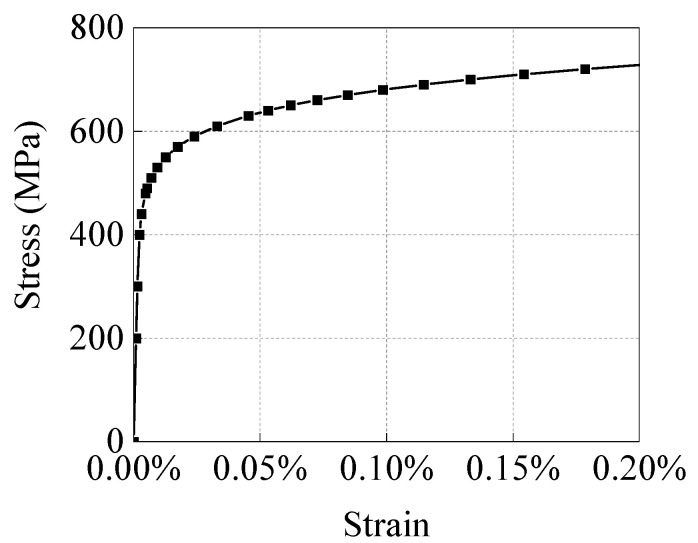
True stress–strain curve of X65 pipeline steel.

**Figure 9 materials-15-05492-f009:**
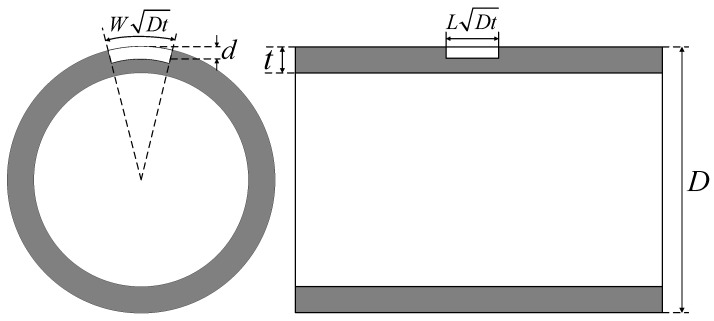
Schematic diagram of external corrosion defect.

**Figure 10 materials-15-05492-f010:**
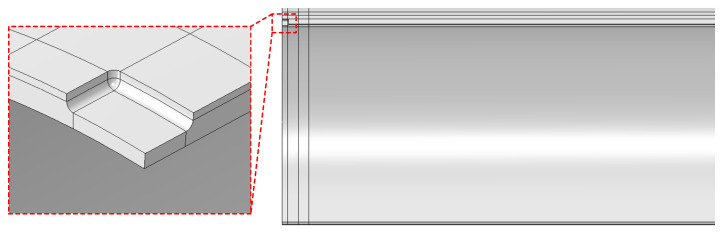
Corrosion interior edge chamfer.

**Figure 11 materials-15-05492-f011:**
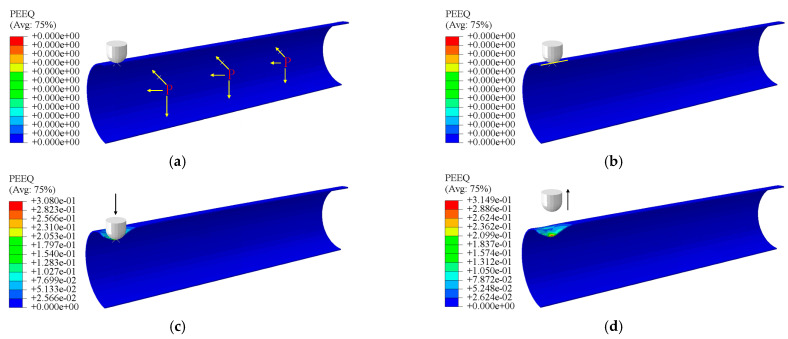
Analysis of steps setting: (**a**) internal pressure load, (**b**) setting the contact, (**c**) the displacement load, (**d**) the rebound of the dent.

**Figure 12 materials-15-05492-f012:**
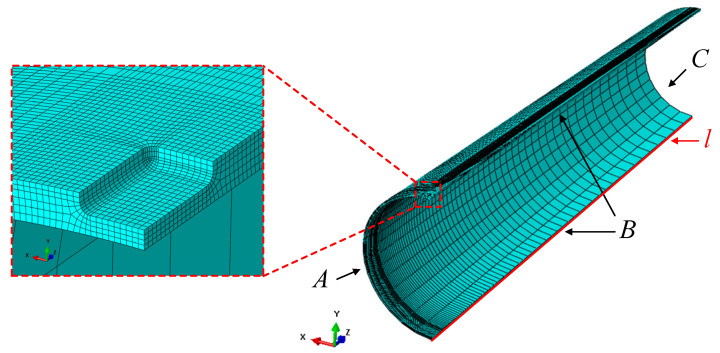
Boundary condition and mesh generation.

**Figure 13 materials-15-05492-f013:**
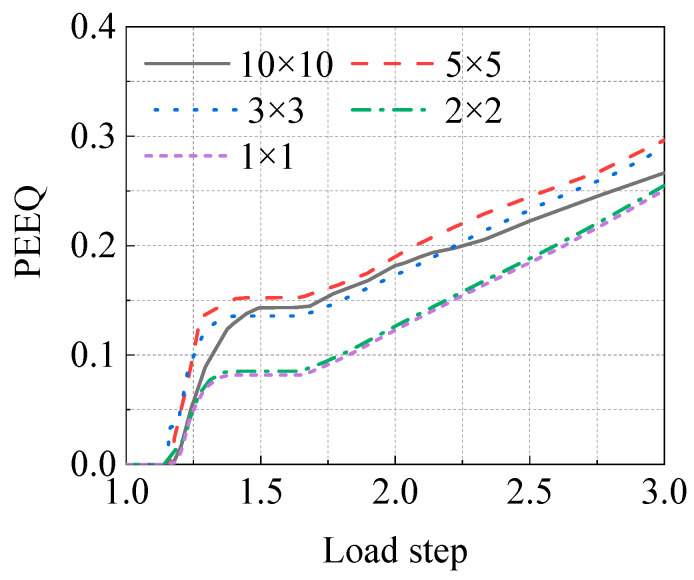
PEEQ comparison of center point of the internal surface with different mesh sizes.

**Figure 14 materials-15-05492-f014:**
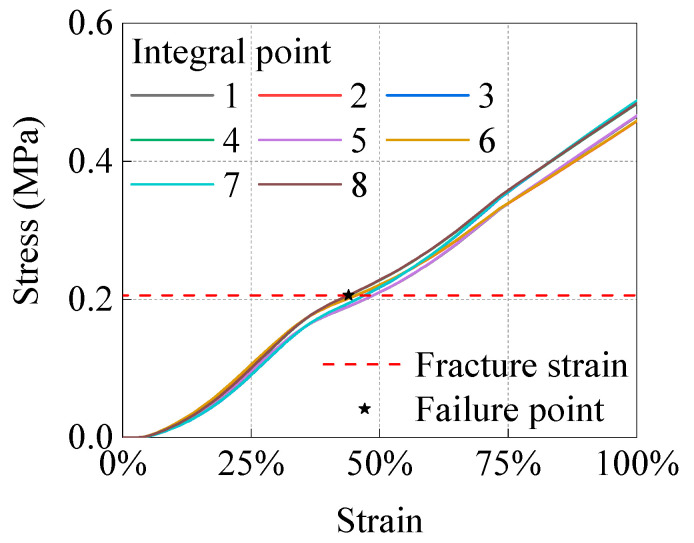
Changes in PEEQ at the integration point of the first failure element with dent depth.

**Figure 15 materials-15-05492-f015:**
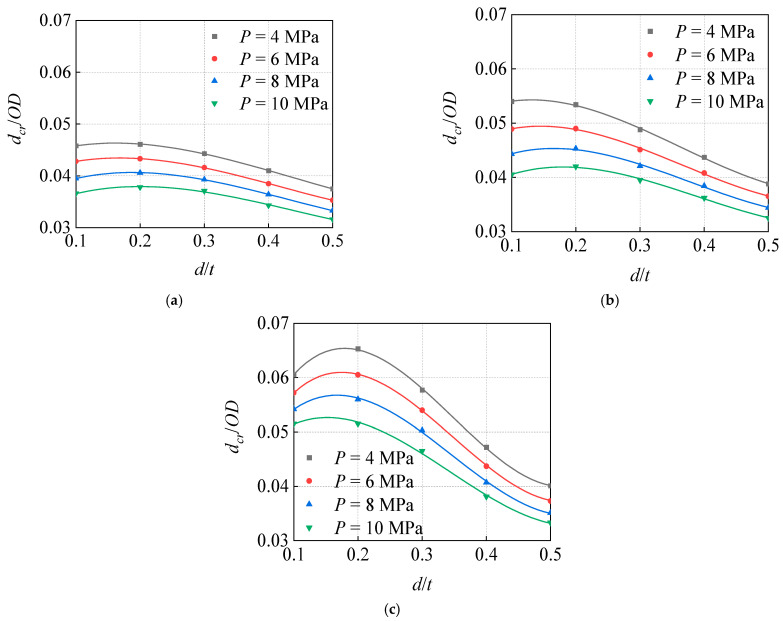
Influence of corrosion depth on critical dent depth under different pressure: (**a**) *R*/*OD* = 0.08, (**b**) *R*/*OD* = 0.10, (**c**) *R*/*OD* = 0.12.

**Figure 16 materials-15-05492-f016:**
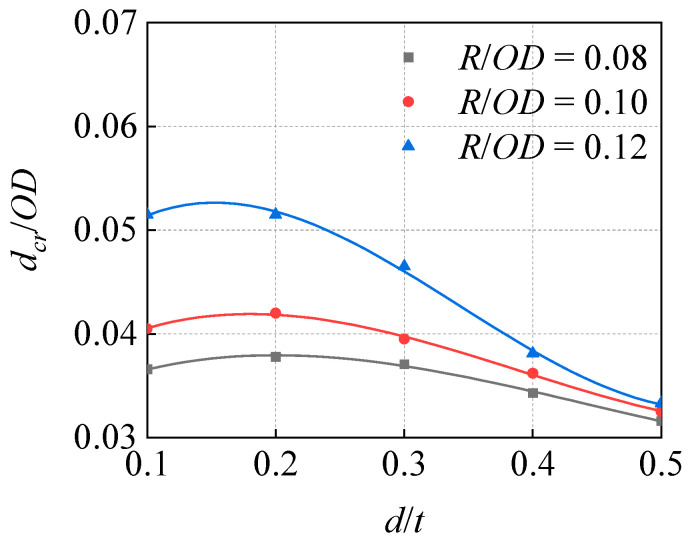
Influence of corrosion depth on critical dent depth under indenters with different curvature radii (*P* = 10 MPa).

**Figure 17 materials-15-05492-f017:**
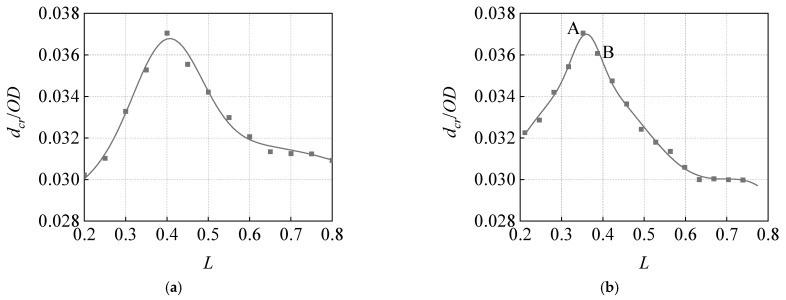
Influence of corrosion length and width on critical dent depth: (**a**) corrosion axial length, (**b**) corrosion circumferential width.

**Figure 18 materials-15-05492-f018:**
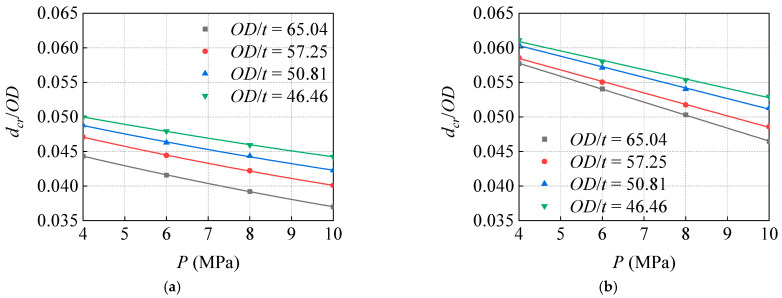
Influence of pressure on critical dent depth under different diameter–thickness ratio: (**a**) *R*/*OD* = 0.08, (**b**) *R*/*OD* = 0.12.

**Figure 19 materials-15-05492-f019:**
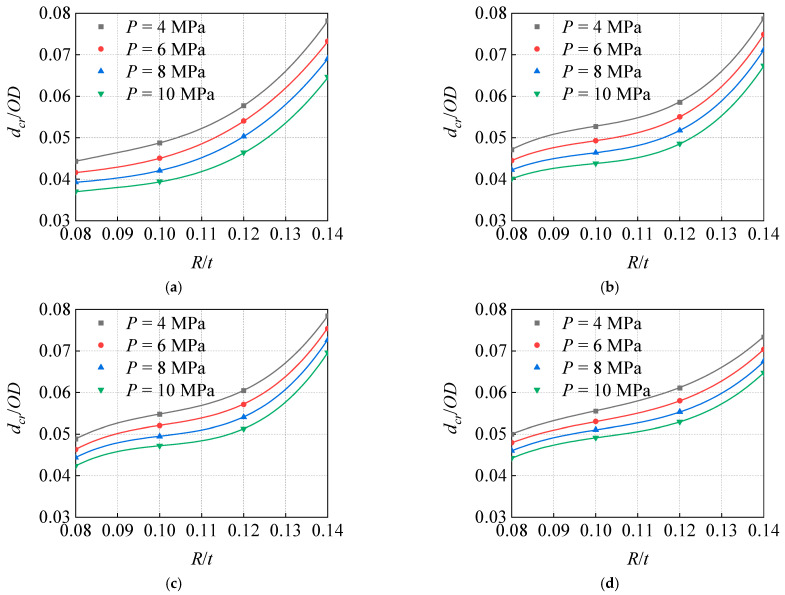
Influence of indenter radius on critical dent depths under different pressure: (**a**) *OD*/*t* = 65.04, (**b**) *OD*/*t* = 57.25, (**c**) *OD*/*t* = 50.81, (**d**) *OD*/*t* = 46.46.

**Figure 20 materials-15-05492-f020:**
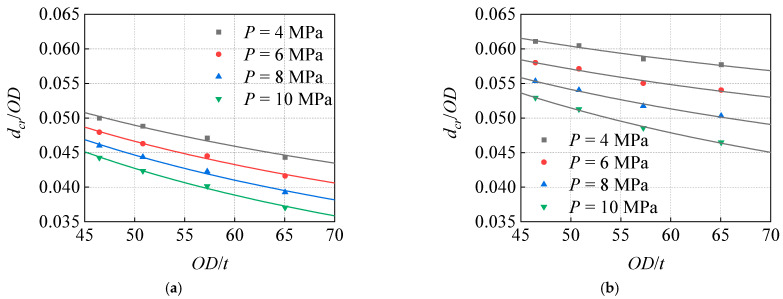
Influence of diameter–thickness ratio on critical dent depths under different pressures: (**a**) *R*/*OD* = 0.08, (**b**) *R*/*OD* = 0.12.

**Figure 21 materials-15-05492-f021:**
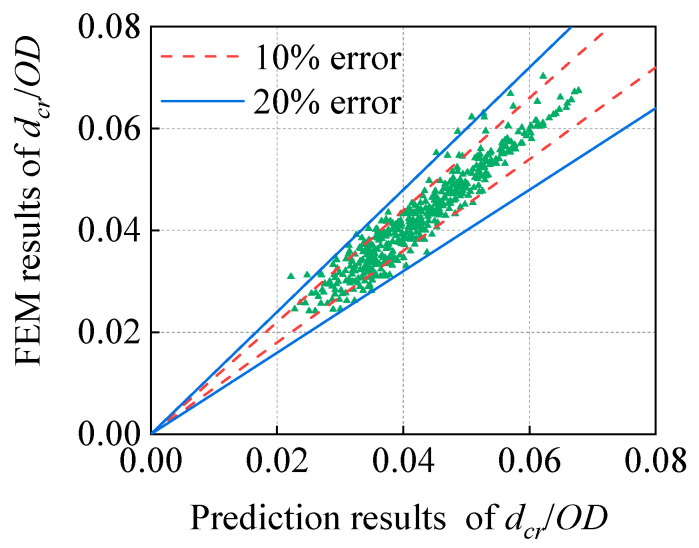
Comparison between the prediction formula and FEM.

**Table 1 materials-15-05492-t001:** Mechanical parameters of pipe for testing.

Material	Elastic Modulus *E* (GPa)	Poisson’s Ratio *μ*	Yield Strength *σ_s_* (MPa)	Tensile Strength *σ_b_* (MPa)
X52	200	0.3	410	498

**Table 2 materials-15-05492-t002:** X65 mechanical parameters of X65 pipeline steel.

Material	Elastic Modulus *E* (GPa)	Poisson’s Ratio *μ*	Yield Strength *σ_s_* (MPa)	Tensile Strength *σ_b_* (MPa)	Elongation	Yield Offset *α*	Strain-Hardening Parameter *n*
X65	200	0.3	551	618	0.233	3.686	11.64

**Table 3 materials-15-05492-t003:** Different mesh sizes and amounts for mesh sensitivity analysis.

Mesh Size in the Local Region of the Corrosion Defect Location		Total Number of Model Meshes
The Axial Length of the Mesh (mm)	The Circumferential Length of the Mesh (mm)	
10	10	22,034
5	5	35,901
3	3	61,242
2	2	102,132
1	1	280,184

**Table 4 materials-15-05492-t004:** Model parameters for analysis of the influence of geometrical parameters on dent depth under different conditions.

Modeling Parameters	Values
Diameter *OD* (mm)	813
Wall thickness *t* (mm)	12.5
Diameter–thickness ratio *OD*/*t*	65.04
Internal pressure *P* (MPa)	4/6/8/10
Curvature radius of indenter *R* (mm)	65.04/81.3/97.56
Ratio of the curvature radius of indenter to diameter *R*/*OD*	0.08/0.10/0.12
Corrosion length coefficient *L*	0.4
Corrosion width coefficient *W*	0.352
Corrosion depth *d* (mm)	1.25/2.5/3.75/5/0.625
Ratio of corrosion depth to wall thickness *d*/*t*	0.1/0.2/0.3/0.4/0.5

**Table 5 materials-15-05492-t005:** Model parameters for analysis of influence of load parameters on dent depth under different conditions.

Modeling Parameters	Values
Diameter *OD* (mm)	813
Wall thickness *t* (mm)	12.5/14.2/16.0/17.5
Diameter–thickness ratio *OD*/*t*	65.04/57.25/50.81/46.46
Internal pressure *P* (MPa)	4/6/8/10
Curvature radius of indenter *R* (mm)	65.04/81.3/97.56/113.82
Ratio of the curvature radius of indenter to diameter *R*/*OD*	0.08/0.10/0.12/0.14
Corrosion length coefficient *L*	0.4
Corrosion width coefficient *W*	0.352
Corrosion depth *d* (mm)	3.75/4.26/4.8/5.25
Ratio of corrosion depth to wall thickness *d*/*t*	0.3

**Table 6 materials-15-05492-t006:** Fitting results of parameters.

Undetermined Parameter	Fitting Result	Undetermined Parameter	Fitting Result
*a* _1_	−7.0005	*b* _4_	0.4732
*b* _1_	10.7404	*c* _4_	0.5196
*c* _1_	−5.5152	*d* _4_	−1.0034
*d* _1_	−1.0185	*a* _5_	48.7906
*e* _1_	−1.0394	*b* _5_	2.7162
*a* _2_	−2.6085	*c* _5_	0.059
*b* _2_	1.5749	*d* _5_	0.3312
*c* _2_	1.4732	*a* _6_	−2.8819
*a* _3_	−1.6493	*b* _6_	−0.6212
*b* _3_	1.3136	*c* _6_	0.4484
*c* _3_	0.7091	Correlation coefficient	0.9557
*a* _4_	−2.2019		

**Table 7 materials-15-05492-t007:** Applicable range for dimensionless parameters.

Dimensionless Parameters	Value Ranges	
	≥	≤
*d*	0.1	0.5
*L*	0.2	0.8
*W*	0.21	0.74
*P*/*P_y_*	0.17	0.59
*R*	0.08	0.14
*OD/t*	46.46	65.04

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
