# Peer review of "A Semi Empirical Regression Model for Critical Dent Depth of Externally Corroded X65 Gas Pipeline"

_materials, 2022, doi:10.3390/ma15165492_

Round 1

Reviewer 1 Report

In the present work “a semi-empirical regression model for critical dent depth of externally corroded X65 gas pipeline” has been investigated. The following comments must be considered before publication:

1.     The authors should confirm the reliability of their modeling method with experimental methods or some references.

2.     The introduction section should be summarized.

3.     There are many writing errors.

Reviewer 2 Report

Congratulations. The work is good, however, there are some concerns about your work, which can be resolved, improving the work and your understanding. Please see the attachment.

Reviewer 3 Report

With the title of “A semi empirical regression model for critical dent depth of externally corroded X65 gas pipeline”, the authors make an interesting and nice study to predict the behaviour of a corrosion dent in a gas pipeline due to inherent changes in internal pressure. The work seems to be well argued and well done due to the good correspondence between the final results showed.

After reading the work I must write the following comments.

1)      In my opinion the authors must review all the work carefully. There are too many errors in the figure’s numbering. There are too many mistakes like: Error! Reference source not found. Sometimes it is very difficult to understand that on the text and the corresponding figure. I am not sure if all the figures are explained into the text.

2)      All the acronyms used must be described, at least, the first time that they appears (OD and SYMS are an example of this).

3)      I think that the extra-information (line 130) about Rafi “a scholar from University of Windsor”, it is not necessary, so such information can be avoided. If not, I think that a brief description for the rest of the authors cited in the bibliography should be added.

4)      The real stress and the real strain cited in the work are more known as true stress and true strain (just in concordance with the symbols used by the authors).

5)      Fig. 3, page 5: review the concordance between the footer of the figure and its correlation with the proper figure.

6)      Have the authors any explanation in relation to the increase of dcr/OD in Figs. 15 and 15 (pages 13-14) between the values of 0.1-0.2 for d/t.

Kind regards.

Reviewer 4 Report

I recommend this paper to be published pending major revision.

(1)    I did not see the convergence test of any quantity of finite element analysis with total number of elements until the results is settle down.

(2)    The author talked about the experimental setup but I did not see any real comparison with the numerical simulation results.

(3)    In the text the author mentioned plastic strain. Is the results reach the yield point? Please explain.

(4)    The author talked about the contact. How does the author do the contact using finite element analysis?

(5)    Which software of finite element analysis did the author use? Is it ANSYS software? Please explain.

Round 2

Reviewer 3 Report

Nothing to add to my previous comments. All of then was solved.

Congratulations to authors for their job.

Best regards.

Reviewer 4 Report

Now the paper is acceptable.